# Ricin Antibodies’ Neutralizing Capacity against Different Ricin Isoforms and Cultivars

**DOI:** 10.3390/toxins13020100

**Published:** 2021-01-29

**Authors:** Maria Lucia Orsini Delgado, Arnaud Avril, Julie Prigent, Julie Dano, Audrey Rouaix, Sylvia Worbs, Brigitte G. Dorner, Clémence Rougeaux, François Becher, François Fenaille, Sandrine Livet, Hervé Volland, Jean-Nicolas Tournier, Stéphanie Simon

**Affiliations:** 1Paris-Saclay University, CEA, INRAE, Medicines and Healthcare Technologies Department (DMTS), SPI, 91191 Gif-sur-Yvette, France; j.prigent@hifibio.com (J.P.); julie.dano@cea.fr (J.D.); audrey.rouaix@cea.fr (A.R.); francois.becher@cea.fr (F.B.); francois.fenaille@cea.fr (F.F.); sandrine.livet@gmail.com (S.L.); herve.volland@cea.fr (H.V.); 2Microbiology and Infectious Diseases Department, Anti-Infectious Biotherapies and Immunity Unit, Army Biomedical Research Institute, 91220 Brétigny-sur-Orge, France; arnaud2.avril@intradef.gouv.fr (A.A.); clemence.rougeaux@intradef.gouv.fr (C.R.); jean-nicolas.tournier@intradef.gouv.fr (J.-N.T.); 3Biological Toxins, Centre for Biological Threats and Special Pathogens, Robert Koch Institute (RKI), 13353 Berlin, Germany; worbss@rki.de (S.W.); dornerb@rki.de (B.G.D.)

**Keywords:** ricin, neutralizing antibodies, recombinant antibodies, monoclonal antibodies, ricin isoforms, *Ricinus communis* cultivars, pulmonary intoxication, treatment, mouse model

## Abstract

Ricin, a highly toxic protein from *Ricinus communis*, is considered a potential biowarfare agent. Despite the many data available, no specific treatment has yet been approved. Due to their ability to provide immediate protection, antibodies (Abs) are an approach of choice. However, their high specificity might compromise their capacity to protect against the different ricin isoforms (D and E) found in the different cultivars. In previous work, we have shown the neutralizing potential of different Abs (43RCA-G1 (anti ricin A-chain) and RB34 and RB37 (anti ricin B-chain)) against ricin D. In this study, we evaluated their protective capacity against both ricin isoforms. We show that: (i) RB34 and RB37 recognize exclusively ricin D, whereas 43RCA-G1 recognizes both isoforms, (ii) their neutralizing capacity in vitro varies depending on the cultivar, and (iii) there is a synergistic effect when combining RB34 and 43RCA-G1. This effect is also demonstrated in vivo in a mouse model of intranasal intoxication with ricin D/E (1:1), where approximately 60% and 40% of mice treated 0 and 6 h after intoxication, respectively, are protected. Our results highlight the importance of evaluating the effectiveness of the Abs against different ricin isoforms to identify the treatment with the broadest spectrum neutralizing effect.

## 1. Introduction

Ricin toxin is a protein with highly toxic properties present in abundance in the seeds of Castor bean plant (*Ricinus communis*). It is a 64 kDa heterodimer, composed of a lectin subunit (ricin toxin B-chain, RTB) that binds to galactosyl residues at the cell surface [1] inducing the internalization of the heterodimer, and a subunit with rRNA N-glycosidase activity (ricin toxin A-chain, RTA) that inactivates the peptide elongation process in eukaryotic cells by depurination of the rRNA 28S in the 60S ribosome subunit [2,3], and thus belongs to the ribosome-inactivating protein (RIP) family. Castor bean plant is naturally found worldwide and cultivated for the production of castor oil for different industries. It is also used as an ornamental plant as well as in the agricultural industry having caused accidental poisoning of animals and humans [4,5]. The wide availability of the plant together with the relative ease of preparation of the crude extract, has made ricin a potential agent of bioterrorism. Additionally, ricin has been explored for potential military use by different nations. Based on these grounds, ricin is classified as a prohibited substance both by the Chemical Weapons Convention (CWC, schedule 1 compound) and the Biological Weapons Convention (BWC) and its possession or purification is strictly regulated and controlled by the Organization for the Prohibition of Chemical Weapons (OPCW) [6,7,8].

Ricin is part of a ricin gene family that encodes up to seven ricin or ricin-like proteins depending on the *R. communis* cultivar (cv.). Ricin isoforms D and E, and the agglutinin RCA120, are the most abundant, with differences in their sequence and toxicity [9,10,11]. The ricin D isoform is present in most *R. communis* cultivars together with the ricin E isoform. RCA120 is a tetramer composed of two ricin-like heterodimers: The RTA sequence has 93% homology with that of ricin D, while the RTB sequence shows 84% homology [12]. These differences make the RCA120 much less toxic than ricin, and confer on it hemagglutination properties. Ricin E, on the other hand, has an RTA identical to ricin D, but its RTB is a hybrid between the RTBs of ricin D and RCA120 [13,14]. Its toxicity and mechanism of action are, however, similar to those of ricin D [15]. While ricin D is present in all castor bean seeds, ricin E is present in most but not all, typically in small grain varieties [16]. For instance, ricin E is not found in *R. communis* cv. zanzibarensis, which is known to contain only the ricin D isoform [17], but it is found in *R. communis* cv. carmencita [10,18]. Moreover, differences in posttranslational modifications (e.g., N-glycosylation), which can also impact toxicity, have been described depending on the cultivar and origin of the seed from which ricin is isolated [18,19].

Depending on the route of exposure to ricin (oral, respiratory or parenteral), the toxicity and clinical signs vary. Inhalation and injection 50% lethal doses (LD_50_) are an estimated 1 to 25 µg/kg in mice, while the oral LD_50_ is an estimated 1.78–30 mg/kg [4]. In all cases, local clinical signs appear within a few hours after poisoning, followed by more general clinical signs, circulatory failure, and death within a few days [7,20,21]. Several therapeutic approaches have been evaluated, though none has been clinically approved to date: Polyclonal, monoclonal and recombinant antibodies (Abs), toxin inhibitors (directed against either the toxin itself or cell trafficking), and immunomodulatory drugs, etc. Among these approaches, passive immunity is the most effective and promising [8,22]. However, to our knowledge, their effectiveness has been evaluated using a single ricin source which in most cases was not clearly specified.

The importance of the source of ricin for the implementation of standards has been addressed by the EQuATox consortium [10]. In previous articles, we have shown the protective capacity in vitro and in vivo of a humanized recombinant antibody (rAb) directed against RTA (43RCA-G1) [23] and two mouse monoclonal antibodies (mAbs) directed against RTB (RB34 and RB37) [24]. In both cases, we used a single source of ricin to evaluate their neutralizing capacity, containing only ricin D. In order to be confident in the therapeutic efficacy of Abs, in particular mAbs recognizing a single epitope at the surface of the antigen, it seemed necessary to evaluate their protective capacity not only for ricin D, but also for ricin E, and more generally for ricin contained in different cultivars. To this end, we have evaluated and compared the three Abs (43RCA-G1, RB34, and RB37), using in vitro assays to determine their affinity for different ricin toxin isoforms and their protective capacity both in vitro against different ricin isoforms from different sources and in vivo against a mixture of ricins D and E. We found that the combination of the different Abs prevents cell death in vitro against ricin isolated from different *R. communis* cultivars, and protects mice previously exposed to an equimolar solution of ricin D and E purified from the cv. carmencita, better than each Ab used alone.

## 2. Results

### 2.1. Assessment of In Vitro Cytotoxicity of Different R. Communis Cultivars and Ricin Isoforms

As explained above, depending on the source of ricin, different isoforms and/or glycoforms of the toxin can be found, which might affect their toxicity [18,19]. Therefore, in the first place, we wanted to evaluate and compare the toxicity of different ricin toxin sources in an in vitro cytotoxicity assay. Ricin obtained from *R. communis* cv. carmencita seeds was purified and the two isoforms of ricins D and E were separated. The purity of each isoform was evaluated by liquid chromatography coupled with a Q-Exactive mass spectrometer (LC-MS/MS). Ricin D showed more than 98% purity (having less than 1% contamination with ricin E or RCA120), while ricin E, after several purification processes, reached around 90% purity, having approximately 9% contamination with RCA120 (Appendix A). In order to characterize the activity of our purified isoforms, we compared their in vitro cytotoxicity on cultivated cells with a previous purified ricin D from *R. communis* cv. sanguineus, used in Prigent et al., 2011 and a gift of Dr. Beaumelle (CNRS) [24,25], as well as ricin purified from different *R. communis* cultivars [10].

We evaluated the cytotoxicity of purified ricin D, ricin E, an equimolar mixture of D/E, ricin D “Beaumelle” used as reference and purified ricins from different cultivars, as well as RCA120, using Jurkat cells, and determined the 50% Cytotoxic Dose (CD_50_) values (Figure 1, Table 1).

The results are shown as a percentage of viable cells in comparison to the control cells incubated in the absence of toxin. Ricin isoforms D and E from cv. carmencita and ricin from the different cultivars show a similar cytotoxicity, with a CD_50_ of the same order of magnitude (i.e., 10 pg/mL or 0.16 pM), with some differences (Table 1): As previously described, ricin E is less cytotoxic than ricin D [26], with a CD_50_ which is almost twice that of ricin D (12.8 ± 2.6 pg/mL vs. 7.4 ± 1.0 pg/mL, *p* < 0.001), the cytotoxicity of the equimolar solution of ricins D and E (ricin D+E) being in between (8.8 ± 0.8 pg/mL), though only a statistically significant difference was observed with ricin E (*p* = 0.0078). When compared to the other cultivars, some of them, including zanzibarensis, which is known to produce only ricin D [17], showed a cytotoxicity lower though not statistically significant to that of our purified ricin D (Table 1), while other cultivars, previously shown to produce both isoforms, such as carmencita, have a cytotoxicity similar to that of ricin D+E (Table 1). The cytotoxicity of ricin D from *R. communis* cv. sanguineus (“Beaumelle”) is similar to that of the ricin D/E mixture, rather than to ricin D from cv. carmencita or zanzibarensis, though no statistically significant difference was observed between the CD_50_ values. It should be noted that ricin purified from cv. carmencita at RKI, which naturally contains a 1:1 ratio of isoforms D and E, presents the same CD_50_ as the reconstituted D/E mixture from separately purified ricins D and E at the CEA. For other cultivars, containing different proportions of isoforms D and E, cytotoxicities vary from a CD_50_ close to that of ricin D alone (cv. tanzania, 7.5 ± 1.3 pg/mL) to a CD_50_ close to that of ricin E alone (cv. sanguineus 13.4 ± 0.7 pg/mL). These differences might correspond to different proportions of ricin D vs. ricin E or differences in activity due to posttranslational modifications and/or purification process. As expected, RCA120 cytotoxicity was at least 10 times lower than that of the different ricins (209.0 ± 26.3 pg/mL, *p* < 0.001).

### 2.2. Kinetic Parameters of Ab Interaction with Ricin Isoforms D and E

In order to assess the capacity of our Abs to recognize different ricin isoforms and cultivars and their neutralizing coverage, we evaluated their affinity and in vitro neutralizing capacity using the cytotoxicity assay. To determine the kinetic affinity parameters, we used a fixed concentration of Abs, and serial dilutions of either ricin D or E, and determined the kinetic parameters of association and dissociation by the biolayer interferometry method (Octet^®^ Red96e) (Figure 2 and Table 2).

For ricin D, all the curves show a rapid increase of the signal during the association phase, which indicates a fast association of the Abs (Figure 2a–c), followed by a very slow dissociation phase (second part of the curves) indicating a high stability of the Ab/ricin complexes. Altogether, these results show a high affinity of the Abs for ricin D with an overall K_D_ of 10.0 pM for RB34, 137 pM for RB37, and 48 pM for 43RCA-G1 (Table 2), being RB34 the one with the highest affinity (*p* < 0.001). When evaluating the affinity of the Abs for ricin E, 43RCA-G1 has an affinity for ricin E similar to that for ricin D (K_D_ = 96 pM, Table 2 and Figure 2f), while the two anti-RTB Abs (RB34 and RB37) display a very low affinity for ricin E (Table 2, Figure 2d-e) statistically different from 43RCA-G1/ricin E (*p* < 0.001). RB34 has an association rate for ricin E similar to that for ricin D, but a high dissociation rate (Figure 2d), suggesting that the complexes RB34/ricin E are much less stable than the RB34/ricin D ones, giving an overall affinity constant of 5.42 nM, which is 400-fold less than that for ricin D (*p* < 0.001). Concerning RB37, we were unable to calculate a dissociation constant since no association with ricin E was detectable (Figure 2e).

### 2.3. In Vitro Neutralizing Capacity of the Abs

In order to evaluate the capacity of the three Abs to neutralize the ricin isoforms D and E, we used a neutralizing in vitro assay [24]. Jurkat cells were incubated in the presence of a constant concentration of ricin D or E purified from the cv. carmencita and of ricin purified from the different cultivars. The concentration of ricin was determined as the concentration inducing more than 95% mortality and ranged from 0.0625 to 0.250 ng/mL final concentration depending on the ricin isoform or cultivar toxicity (see Table 3). After a 3-day incubation with different Ab concentrations ranging from 0 to 100 µg/mL, we determined the percentage of cell viability in each condition, and the 50% Inhibitory Concentration (IC_50_) for each Ab or combination of Abs.

In Table 4, we show the IC_50_ of Abs or Ab combinations for the different ricin isoforms from *R. communis* cv. carmencita, and for the different *R. communis* cultivars. A statistical analysis was performed (one-way ANOVA with Tukey’s post-hoc test) to compare first the inhibitory capacity of the individual Abs and then the best individual Ab with the different Ab combinations. All the individual Abs inhibit the ricin D activity, RB34 being the most efficient (i.e., lower IC_50,_
*p* < 0.001), which correlates with its kinetic parameters. When combining the Abs in equal amounts, the RB34+43RCA-G1 combination gives the best protection, and the addition of RB37 does not improve the protecting capacity. Concerning ricin E neutralization, 43RCA-G1 is the best candidate, as expected from kinetic parameters. In this case, we confirmed the weak affinity of RB34 for ricin E, since the addition of RB34 to 43RCA-G1 is less effective than 43RCA-G1 alone in the in vitro neutralization of ricin E (*p* < 0.01). Finally, when evaluating the neutralization of the equimolar solution of ricins D and E, we observed that RB34 and 43RCA-G1 have a similar neutralizing capacity, RB37 being much less effective. Moreover, a combination of RB34 and 43RCA-G1 provides a synergistic neutralizing effect, with an IC_50_ 12 to 15 times lower than the IC_50_ of each Ab individually (*p* < 0.001).

Concerning the capacity of the Abs to neutralize ricin purified from different cultivars, in most cases, the combination of RB34 and 43RCA-G1 had the best neutralizing capacity, with an IC_50_ 2 to 12 times lower than that of each Ab individually (*p* < 0.01), except for *R. communis* cv. zanzibarensis, which contains exclusively ricin D [17], and tanzania and impala, for which RB34 showed the best neutralizing capacity, though no statistically significant difference from RB34+43RCA-G1 was observed.

#### Ricin-Neutralizing Mechanism of the Anti-RTB mAbs RB34 and RB37

We have previously shown that RB34 and RB37 recognize RTB. Since it has previously been described that some anti-RTB Abs could compete with the galactose binding sites of RTB to inhibit the ricin entry into cells [27,28,29], we previously performed a competition assay using RTB, anti-RTB Abs, and lactose as competitor, and we have shown that lactose inhibited the binding of RB37 to RTB, while it enhanced the binding of RB34 to RTB [24]. In order to confirm these data and to understand more deeply the relationship between lactose and the binding of the mAbs to ricin, we evaluated their affinity for ricins D and E and RTB in the absence or presence of lactose, by the biolayer interferometry method. In the presence of lactose, the affinity of RB37 for ricin D or RTB is reduced (i.e., K_D_ value increased) due both to a reduction of the association and an increase of the dissociation, giving an overall K_D_ around 10 times higher (*p* < 0.001) (Table 5), confirming the competition between lactose and RB37 for the binding site of RTB, whereas the affinity of RB34 for ricin D increases by around five times in the presence of lactose (*p* < 0.001), suggesting that the binding of ricin to its ligand induces a conformational change in holo-ricin D that enhances the binding, yielding a more stable RB34/ricin D complex (Table 5).

Since RB34 can bind to ricin in the presence of lactose, while RB37 acts in competition with lactose in binding to ricin, we hypothesized that RB37, unlike RB34, should prevent the binding (and thus internalization) of ricin to the cells. To verify this hypothesis, we evaluated first the binding of fluorescent ricin D to the surface of Vero cells in the presence of mAb RB34 or RB37 by incubation at 4 °C and secondly its internalization by incubation of the cells at 37 °C. A non-neutralizing anti-RTB mAb (RB22) was used as control. The binding of ricin and consequently its internalization into the cells is blocked by RB37 (Figure 3, panels c and h) and by lactose (Figure 3, panels d and i). Neither the binding (Figure 3, panels b and e) nor the internalization (Figure 3, panels g and j) of ricin is prevented by RB34 or RB22, with a labeling suggesting an intracellular localization unmodified in comparison to ricin alone (in the endoplasmic reticulum or Golgi apparatus) and suggesting that the neutralizing capacity of RB34 is most probably associated with a blockage at some point between the retrograde transport and the release of the RTA.

It should be noted that it has been previously shown that the neutralizing capacity of 43RCA-G1, directed against RTA is associated with the inhibition of the enzymatic activity of ricin, as demonstrated by cell-free assays consisting of a cell-free translation system which is put in contact with ricin to evaluate the inhibition of translation [30].

### 2.4. In Vivo Protective Capacity of the Abs

Based on the results obtained in vitro, we chose the two best Abs (RB34 and 43RCA-G1) to evaluate their neutralizing capacities in vivo in a murine model, using a mixture of isoforms D and E of ricin. In order to evaluate the time-window of action of the neutralizing Abs in prophylaxis, we first assessed the half-lives of the Abs in mice.

#### 2.4.1. Pharmacokinetic Studies of the Abs RB34 and 43RCA-G1

In previously published experiments [24], we evaluated the pharmacokinetic (PK) parameters of RB34 in CD1 mice. These parameters show a fast transfer of RB34 to the bloodstream (peak of the curve 18–24 h post-injection) and a low rate of elimination. In order to verify if the time window of action was similar in BALB/cJ mice and to evaluate the PK parameters of the humanized 43RCA-G1 and to compare both in the same experimental conditions, we measured by immunoassay the Ab concentration in blood samples at different time points after the intraperitoneal (i.p.) administration.

Both Abs are detected in the blood of the mice a few hours after i.p. injection, reaching the peak in the blood compartment between 24 and 48 h post-injection, with a maximum concentration of RB34 greater than that of 43RCA-G1 (respectively close to 30 and 20 µg/mL) (Figure 4).

A non-compartmental PK analysis allowed us to determine the half-life of RB34 and 43RCA-G1 as 12 and 4 days, respectively. Due to the difference in the nature of these Abs, the difference observed in their half-lives is in accordance with what we expected and in line with published data [31]. However, since the clinical signs of poisoning are observed around 24 h after ricin exposure, and mortality in our mouse model occurs between days 3 and 5, the window of action of both Abs seems adequate.

#### 2.4.2. In Vivo Protective Effect of the Abs RB34 and 43RCA-G1, Separately or in Combination

LD_50_ of ricin D+E (1:1) administered by the intranasal (i.n.) route was first determined in BALB/cJ female mice. The mice were instilled intranasally with different ricin doses and their survival was recorded for 21 days. The LD_50_ of ricin D+E was calculated by the Miller and Tainter method [32] at 20 µg/kg (Appendix A).

For the in vivo protection assay, mice were poisoned by the i.n. route with 5 LD_50_ of ricin D+E (from cv. carmencita) and treated concomitantly, 6 or 24 h later by i.p. administration of the Abs individually or in combination at a final dose of 10 mg/kg. When the treatment was administered at the same time as ricin poisoning, the combination of RB34 and 43RCA-G1 conferred 55% protection, 43RCA-G1 alone conferred 20% protection, and no protection was obtained with RB34 alone (Figure 5a). When administered later (6 h after poisoning), 42% protection was provided by the combined RB34+43RCA-G1 administration (Figure 5b), and this fell to a 10% protection when administered 24 h after poisoning (Figure 5c). These results are in line with those observed in vitro, the combination of Abs being the best therapeutic option.

## 3. Discussion

Ricin poisoning continues to be a major biothreat and is classified by the CDC as a category B substance [6]. Indeed, real incidents have been prevented in several countries [33]. Despite the efforts made in the last decades to find a treatment, none has been clinically approved yet. In 2017, Gal et al. reviewed the different studies of treatments of pulmonary ricinosis, and the most promising treatment was passive immunotherapy [22]. The advantages of using mAbs or rAbs are that they can be highly specific and have reduced immunogenicity compared to polyclonal Abs, and also their toxicity can be reduced by molecular engineering (chimerization, humanization) [23,30,31,34,35,36]. Moreover, they have a long half-life and can be used for prophylactic or therapeutic purposes [37,38]. Several studies have evaluated the protective capacity of different mAbs directed against RTA or RTB, and different strategies have been proposed to extend the therapeutic window or the efficacy, such as a combination of Abs targeting different epitopes of the different ricin subunits [24,39,40] or a combination with other agents such as doxycycline to reduce inflammation [41]. Depending on the origin of the ricin beans, the ricin amounts and composition might vary, due to differences in ricin D or E proportions, and/or posttranslational modifications, which might affect their toxicity and their recognition by Abs [4,19,42]. However, to the best of our knowledge, in all the studies, the neutralizing capacities of the Abs were tested only against a single cultivar and/or a single isoform of ricin, which in most cases was not specified. This point was highlighted in only one article where a cocktail of high-affinity mAbs directed against different epitopes was used and showed a very high protective capacity even 48–72 h after poisoning. Although mentioned as a perspective, an evaluation of this cocktail in order to provide a broad-spectrum protection against different cultivars has not been published yet [39]. In this study, we addressed this issue using different techniques to evaluate the affinity and neutralizing capacity of our Abs against different ricin isoforms and ricin extracted from different cultivars.

The correlation between the kinetic parameters of ricin/Ab interaction, particularly the k_off_ (or k_dis_) value, and the neutralizing capacity of the Abs has been demonstrated [43,44]. Here, we observed that RB34 has a very high affinity, due particularly to a very low k_dis_, for ricin D, but a much lower affinity (400-fold less) for ricin E, due mainly to a faster (300-fold) dissociation. When evaluating the neutralizing capacity of RB34 in vitro, we observed a good neutralizing capacity against ricin D, and against *R. communis* cv. zanzibarensis, which is known to contain only ricin D, but the mAb was less efficient against cultivars containing both ricin D and E isoforms and the protection of cultivated cells against ricin E toxicity was completely abolished. Regarding 43RCA-G1, since both ricin D and E isoforms share the same RTA, it has a similar affinity for ricins D and E, and shows a similar in vitro neutralizing capacity against the different cultivars or isolated isoforms, though it was more effective in neutralizing ricin E. Finally, the inability of RB37 to bind to ricin E makes it impossible to neutralize its toxicity, whereas it is effective in protecting cells from the cytotoxicity of ricin D isolated from *R. communis* cv. carmencita, or ricin from the cv. zanzibarensis. The combination of RB34 and 43RCA-G1 showed a synergistic effect in vitro, in most cases the IC_50_ of the equimolar combination of RB34 and 43RCA-G1 being 10-fold lower than that of each Ab alone.

The effectiveness of mAbs RB34 and RB37 has been previously evaluated with ricin D from the cv. sanguineus and showed 90% and 100% protection, respectively when administered by the intravenous (i.v.) route at 5 mg/kg (approximately 100 µg/mouse) 1 h after administration by the i.n. route. Moreover, the combination of these Abs and an anti-RTA mAb (RA36) protected 90% of mice when administered up to 7.5 h after poisoning [24]. For 43RCA-G1, a full protection was observed when administered by the oropharyngeal route (50 µg/mouse) 6 h after poisoning by the same route with ricin at 16 µg/kg, and 60% protection when administered 24 h after poisoning [23]. Here, we compared the therapeutic capacity of RB34 and 43RCA-G1 (alone or combined) when administered by the i.p. route to mice previously poisoned with an equimolar mixture of ricins D and E from the cv. carmencita by the i.n. route. As has been observed in the cytotoxicity assay, the synergistic effect of the combination of both Abs gave 55% and 42% protection when administered 0 and 6 h after poisoning, respectively, but only 20% and 15% of mice survived when treated with 43RCA-G1 alone at the same time points and no protection was observed with RB34. These differences in comparison to our previous published studies can be explained by the type of ricin used, which was less favorable for RB34, which binds efficiently only half of the administered toxin, and by the routes of administration of the Abs (i.v. in previous studies vs. i.p. in this study). These results highlight the importance of evaluating the efficacy of Abs against different ricin isoforms and cultivars, and the route of countermeasure administration, which appears to be more effective when local than systemic (for 43RCA-G1) and more effective when i.v. than i.p. (for RB34). This is probably due to a faster access to the site of poisoning (lung) when administered locally (43RCA-G1) or directly in the blood compartment (RB34), compared to the i.p. route. The local and/or i.v. routes of administration of the Ab combination will certainly need to be evaluated further in the future, as will combinations with molecules with complementary mechanisms of action, such as disease-modifying drugs or small molecules known to inhibit intracellular trafficking [41,45,46].

Concerning the mechanism of action of the Abs, we know from published data that 43RCA-G1 inhibits the enzymatic activity of RTA [30]. For RB34 and RB37, we previously showed by competitive enzyme-linked immunosorbent assay (ELISA) that RB37 competes with galactose in binding to RTB, suggesting the inhibition of the binding of ricin to the cells, while the binding of RB34 to RTB was enhanced with galactose, suggesting a completely different mechanism of inhibition [24]. To take the understanding of the mechanism of action of the RB34 and RB37 mAbs one step further, we measured the association and dissociation rates of RB34 and RB37 with ricin D in the presence of lactose and confirmed that RB37 acts as a competitive inhibitor of ricin D or RTB and once bound blocks the further binding of lactose. This was confirmed by immunofluorescence, where we demonstrated that RB37 inhibits the binding of ricin to the cells and thus its internalization. Conversely, the affinity of RB34 for ricin D increased almost 5-fold as a result of a combined increase of the association rate constant and decrease of the dissociation rate constant. The fact that RB34 does not inhibit the binding of ricin to cells or its internalization into the cells strongly suggests that its neutralizing capacity is most probably associated with a blockage inside the cell at some point between the internalization, retrograde transport, and the release of the RTA into the cytoplasm, as previously described for other anti-RTB Abs [47]. The determination of the epitope recognized by RB34 and a deep understanding of its mechanism of action would shed new light on host proteins involved in the intracellular trafficking of ricin.

## 4. Materials and Methods

### 4.1. Ricin Extracts

Ricin toxin from *R. communis* cv. carmencita seeds was purified as previously described [17]. Briefly, decorticated and defatted castor beans were soaked in an acetic acid solution pH 4, homogenized, and incubated overnight at 4 °C for protein extraction. After centrifugation, the supernatant was first dialyzed against distilled water for 48 h and then against 10 mM Tris-HCl, pH 7.4, for 24 h. Ricin was purified by affinity chromatography using a *p*-aminobenzyl 1-thio-β-D-galactopyranoside–Agarose (Sigma-Aldrich) affinity column. After washing, ricin was eluted using a linear gradient from 0 to 250 mM galactose in a 10 mM Tris buffer, pH 7.7. The fractions were analyzed by sodium dodecyl sulfate-polyacrylamide gel electrophoresis (SDS-PAGE), and ricin-containing fractions were pooled. Further fractionation of the ricin toxin to purify ricin isoforms D and E was achieved by ion-exchange chromatography on a strong cation-exchange column 0 to 500 mM NaCl in 10 mM Tris-HCl, pH 7.7. The protein content was determined by absorbance at 280 nm and purity (i.e., residual content of each isoform) was assessed after trypsinolysis by liquid chromatography coupled with a Q-Exactive mass spectrometer (Thermo Fisher Scientific, Courtaboeuf, France) operated in the data-dependent acquisition mode, as previously described [48]. The peak area signal of the following tryptic peptides was used for calculation: NGLCVDVTGEEFFDGNPIQLWPCK, SNTDWNQLWTLR, DSTIR, CLTISK peptides (representative of RCA120); VWLEDCTSEK, DNCLTTDANIK, GTVVK, SDPSLK peptides (representative of RCA120 and ricin E); DNCLTSDSNIR, NDGTILNLYSGLVLDVR, ASDPSLK, QIILYPLHGDPNQIWLPLF peptides (representative of ricin D).

Ricin toxin from the seeds of different *R. communis* cultivars was purified, as previously described [10]. Briefly, the purification process consisted of an acid extraction at pH 4 and an ammonium sulfate precipitation of proteins. Affinity separation using an in-house prepared lactosyl-Sepharose-4B column was used to isolate ricin and RCA120 from other constituents in the precipitate. Finally, size exclusion chromatography over the HiLoad Superdex 200 prep grade (GE Healthcare, Uppsala, Sweden) was used to separate ricin from RCA120. For determination of the concentration of the purified ricin preparations, the absorption at 280, 260, and 320 nm were measured in independent dilutions using a NanoPhotometer (Implen, Munich, Germany). The average of the absorption values of each dilution was used to calculate the concentration of the toxin preparations according to Lambert-Beer’s law with ε (ricin) = 1.1615 mL × mg^−1^ × cm^−1^ [10].

### 4.2. Recombinant and Monoclonal Abs

Mouse mAbs RB34 and RB37 were purified by high-throughput capture chromatography using Protein A resin (ProSep^®^-A, Merk-Millipore, Darmstadt, Germany), eluted with a 0.1 M glycine pH 2.5 buffer, and dialyzed overnight against a 50 mM potassium phosphate buffer, pH 7.4. Their purity and integrity were determined by polyacrylamide gel electrophoresis (SDS-PAGE), using the Agilent Protein 230 Kit© (Agilent Technologies Inc., Santa Clara, CA, USA) in reducing and non-reducing conditions, following the manufacturer’s instructions, and their concentration was determined by UV absorbance at 280 nm (Cary 8454 UV-Vis, Agilent Technologies Inc., Santa Clara, CA, USA).

Humanized rAb 43RCA-G1 was previously produced and purified as described by Respaud et al. [23]. Briefly, 43RCA-G1 was purified by affinity chromatography, using the recombinant protein A-Sepharose fast flow (GE Healthcare, Uppsala, Sweden). A second purification step was further performed using the HiPrep 16/10 SP FF cation exchange column (GE Healthcare, Uppsala, Sweden). The Ab was eluted with 25 mM citrate, pH 3.0, and dialyzed overnight against PBS, pH 7.4, and then concentrated to 15 mg/mL using a centricon 30 kDa MWCO. Ab integrity and purity were verified by the SDS-PAGE analysis (10% precast SDS-PAGE gel, NuPAGENovex, Thermo Fisher Scientific Inc., Illkirch, France), under reducing and non-reducing conditions.

### 4.3. Cell Viability Assays

Cell viability assays were performed as previously described by Prigent et al. [24]. Briefly, Jurkat cells (from ATCC) were cultured in RPMI 1640 without a phenol red medium (GIBCO), supplemented with 10% fetal calf serum (GIBCO), 1% glutamine, 1% sodium pyruvate, 1% non-essential amino acid MEM solution, and 1% penicillin/streptomycin, all from Sigma-Aldrich, at 37 °C in a humidified atmosphere with 5% CO_2_.

For the ricin toxicity tests, different ricin concentrations ranging from 0 to 100 ng/mL (final concentrations) were distributed in 96-well clear flat-bottom polystyrene tc-treated microplates (Corning Inc., Corning, NY, USA), and Jurkat cells were added at a ratio of 1000 cells per well (i.e., 10^4^ cells/mL). Cells were incubated at 37 °C in a humidified atmosphere with 5% CO_2_ for 72 h. After incubation, viability was assessed by the measurement of ATP relative concentrations, using the CellTiterGlo^®^ Luminescent Cell Viability Assay kit (Promega, Madison, WI, USA), following the manufacturer’s instructions. Samples without ricin (cells only) were used to determine the 100% viability, allowing a calculation of the percentage viability for each sample.

To determine the Ab neutralizing capacity, the lowest concentration of ricin killing more than 95% of cells (indicated in Table 3) was used. Purified Abs (concentration ranging from 0 to 100 µg/mL final concentration) were mixed with the toxin in a 50 µL volume and pre-incubated for 60 min at 37 °C in 96-well plates. After the pre-incubation, 50 µL of Jurkat cells at a density of 2 × 10^4^ cells/mL was added to each well (i.e., 1000 cells per well). After 3 days of incubation at 37 °C, viability was assessed as indicated above. The combined effects of neutralizing Abs were evaluated using combinations of two or three Abs in equal proportions and at concentrations equivalent to those used for Abs alone (0 to 100 µg/mL total final concentration).

### 4.4. Determination of Abs/Ricin Kinetic Parameters

The kinetic parameters of the Abs for the different ricin isoforms were determined by biolayer interferometry using the Octet^®^ Red96e from fortéBio^®^ (Sartorius, Fremont, CA, USA) system, following the manufacturer’s instructions. Briefly, anti-mouse (AMC) or anti-human (AHC) Fc capture biosensors from fortéBio^®^ (Sartorius, Fremont, CA, USA), previously hydrated by immersion in an assay buffer (0.1 M phosphate buffer, pH 7.4, with 0.15 M NaCl, 0.1% bovine serum albumin, 1% sodium azide, and 0.02% Tween20) for at least 10 min, were loaded with the Ab of interest by immersion in a solution with an Ab concentration of 5 µg/mL (RB34 and RB37) or 2.5 µg/mL (43RCA-G1) for 300 s, followed by determination of the baseline signal by immersion in an assay buffer for 200 s. The loaded sensors were then immersed for 900 s in different solutions containing different ricin concentrations (one sensor was used for each concentration of the range), ranging from 0 to 5 nM for ricin D and from 0 to 80 nM (RB34 and RB37) or 0 to 10 nM (43RCA-G1) for ricin E, and changes in the signal due to the association of the ricin with the Abs were recorded. The sensors were then washed in an assay buffer for 1800 s, and changes in the signal due to the dissociation of ricin from the Abs were recorded. Data were analyzed using the Octet^®^ Data Analysis HT 10.0 software (fortéBio^®^ LLC, Sartorius, Fremont, CA, USA) and a 1:1 binding model, and the association rate constant (k_a_), dissociation rate constant (k_dis_), and equilibrium dissociation constant (K_D_) were obtained.

To evaluate the effect of lactose on the Ab/ricin D or Ab/RTB interaction, the assay buffer of the association and dissociation steps was supplemented with lactose at a 10 mM final concentration, or with the same molarity of glucose as the control. Ricin D and RTB concentrations ranged from 0 to 5 nM.

### 4.5. Fluorescent-Ricin Cell-Binding or Cell-Internalization Test

Coupling of ricin to the fluorophore FP-532 NHS (FluoProbes^®^, Interchim), was performed by mixing the two compounds in a molar ratio of 1:5, in a 0.1 M borate buffer, pH 9, 0.25 M NaCl. After 1 h of incubation at room temperature, 100 µL of Tris-HCl 0.1 M, pH 8, was added. After 20 min of incubation, the toxin was purified on a G25 column (GE Healthcare, Uppsala, Sweden) and its concentration was measured using the Pierce™ BCA protein assay kit (Thermo Fisher Scientific, Illkirch, France). The cytotoxicity of ricin-FP532 has been evaluated using the cell viability test and was shown to be similar to the unlabeled ricin.

Vero cells were first cultured in a Lab-Tek™ Chamber Slide™ (Thermo Fisher Scientific, Illkirch, France) at 50,000 cells/chamber, for 18 h at 37 °C in a humidified atmosphere with 5% CO_2_. Simultaneously, a ricin-FP532 solution at 1 µg/mL in PBS with 3% BSA, was pre-incubated alone or in the presence of RB34 or RB37 Abs (500 µg/mL), and lactose (50 mM) or RB22 (500 µg/mL) as controls, for 18 h at 37 °C. After pre-incubation, the different solutions were added to the chambers containing Vero cells and incubated either at 4 °C for 30 min to favor ricin binding to the cell surface, or for 4 h at 37 °C to favor ricin internalization. Cells where then washed with PBS-3% BSA and fixed with 4% PFA for 15 min at 4 °C. Fluorescent ricin was localized using an Olympus^®^ IX71 microscope (Olympus Europa GMBH, Hamburg, Germany).

### 4.6. Animal Experimentation

#### 4.6.1. Ethics Statement

All the experiments in mice were performed in accordance with the French and European regulations on the care of laboratory animals (European directives 2010/63/UE, French Law 2001-486, 6 March 2018) and with the agreements of the Ethics Committee of the French Alternative Energies and Atomic Energy Commission (CEtEA, no. 044) following the NRC Guide for the Care and Use of Laboratory Animals. Protocol No. 17-026 delivered to S.S. by the French Veterinary Services and CEA agreement D-91-272-107 from the Veterinary Inspection Department of Essonne (France).

#### 4.6.2. Mice

BALB/cJ female mice aged 8 to 10 weeks from Janvier Labs (France) were housed in ventilated cages, in a room with a controlled environment at 22 ± 2 °C, 50 ± 5% humidity, and 12/12 h dark/light periods. Mice were allowed to habituate to housing conditions for 1 week prior to the beginning of the experiments as recommended. Food and water were administered *ad libitum*, and cages were enriched with cellulose mouse huts and HydroGel^TM^ (BioServices, Uden, Netherlands) to limit the dehydration of the mice.

#### 4.6.3. Pharmacokinetics of Abs

Mice were injected i.p. with a single dose of Abs (50 µg). Blood samples (each group of four mice) were taken at different time points (1, 7, 24, 48, 96, 360, 576, and 1032 h), and collected in tubes containing 100 mM EDTA. After centrifugation (20 min, 4 °C, 15,000 rpm), the plasma was harvested and stored at −20 °C until use. Plasma Ab concentrations were determined by ELISA.

Plasma RB34 concentrations were determined as previously described by Prigent et al. [24]. Briefly, 96-well Nunc™ MaxiSorp Immuno-Plates (Thermo Fisher Scientific, Illkirch, France) were coated with a 100 µL/well of 2 µg/mL ricin D and saturated with an enzyme immunoassay (EIA) buffer (0.1 M phosphate buffer, pH 7.4, with 0.15 M NaCl, 0.1% bovine serum albumin, and 1% sodium azide). Diluted plasma samples from mice that received RB34, or different concentrations of RB34 diluted in the plasma of control mice ranging from 0.135 to 10 ng/mL for the standard curve, were added to the coated plates (100 µL per well), and incubated for 18 h at 4 °C. After a 5-cycle wash (0.01 M PBS, 0.05% Tween20), 100 µL of AChE-labeled goat anti-mouse IgG1, at 2 EU/mL (Ellman units), was added to each well and incubated for 3 h at 25 °C. After a final 5-cycle wash, 200 µL/well of Ellman’s reagent [49] was added and absorbance at 414 nm was measured after 30-min of incubation.

To determine plasma 43RCA-G1 concentrations, 96-well Nunc™ MaxiSorp Immuno-Plates (Thermo Fisher Scientific, Illkirch, France) were coated with 100 µL/well of goat anti-human IgG Fc specific (Sigma-Aldrich), at 5 µg/mL, and saturated with an EIA buffer. Different 43RCA-G1 concentrations ranging from 0.625 to 20 ng/mL diluted in the plasma of control mice for the standard curve, together with the diluted plasma from mice that received 43RCA-G1 by the i.p. route, were added to the previously coated plates (100 µL/well), and incubated for 18 h at 4 °C. After five washing cycles, 100 µL of 50 ng/mL biotinylated ricin D was added to each well and incubated for 1 h at 25 °C. After a five-cycle wash, 100 µL/well of 1 EU/mL AChE-labeled streptavidin was added and the plates were incubated for 30 min at 25 °C. The plates were washed for five cycles, 200 µL/well of Ellman’s reagent was added, and absorbance at 414 nm was measured after 30-min of incubation.

A noncompartmental model and a non-linear one-phase decay fit (PRISM^®^ software v.5.04 GraphPad Software inc., San Diego, CA, USA) were used to determine the half-life of the Abs.

#### 4.6.4. Ricin D+E LD_50_ Determination

To determine the LD_50_ of ricin D+E, three independent experiments were performed. Mice anesthetized with isoflurane (Iso-Vet^®^) were poisoned by i.n. administration of different ricin doses from 7.5 to 150 µg/kg (*n* = 5 to 10 mice following the ethics committee recommendations). The Miller and Tainter method [32] was used to calculate the LD_50_ (data shown in Appendix A).

#### 4.6.5. Antibody Treatment

Mice poisoned by the i.n. route with 5 LD_50_ of ricin D+E (i.e., 100 µg/kg) in a 25 µL volume maximum were treated 0, 6, or 24 h after poisoning by an i.p. administration of 10 mg/kg of RB34, 43RCA-G1, R34+43RCA-G1 (1:1 mix of both Abs), or PBS as the control (*n* = 10 per group) in a 200 µL final volume. Mice were observed twice a day for 21 days and survival was recorded. When required, based on the ethics committee recommendations, a humane endpoint was applied and mice were euthanized.

### 4.7. Statistical Analysis

All cell viability assays, kinetic parameter assays, and in vivo treatment assays were performed at least twice. A statistical analysis for the cell viability assay and in vivo experiments, was performed with the PRISM^®^ software v.5.04 (GraphPad Software Inc., San Diego, CA, USA). For cell viability assays, a non-linear regression 4-parameter variable slope test was used, and one-way ANOVA with Tukey’s post-hoc test was performed. For survival curves of in vivo experiments, log-rank (Mantel-Cox) tests were used. For the kinetic parameter assays, a Student t-test was performed. Data are represented as means ± SEM.

## Figures and Tables

**Figure 1 toxins-13-00100-f001:**
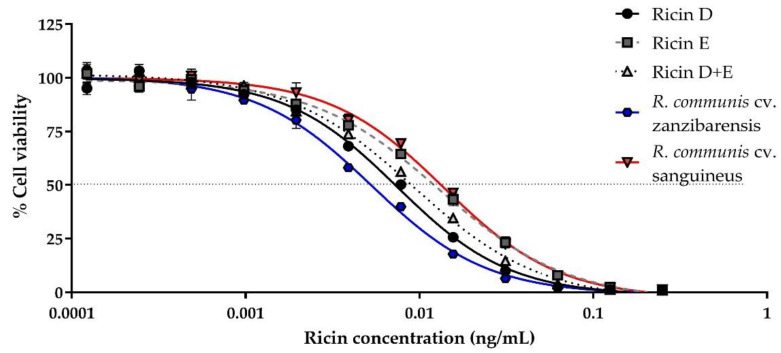
Determination of in vitro cytotoxicity of different ricin isoforms or ricin purified from different cultivars. Jurkat cells were incubated in the presence of different ricin concentrations. Ricin (0–1 ng/mL) was incubated with 10^4^ cells/mL and cell viability was assessed after 72 h by comparison of ATP levels in each well with mean levels of the ATP in control wells (cells without ricin) using the CellTiter-Glo^®^ Luminescence Cell Viability Assay kit (Promega, Madison, United States). Cytotoxicity curves for ricin D (black circles, solid line), E (grey squares, dashed line), or D+E (1:1) (light-grey triangles, dotted line) from *R. communis* cultivar (cv.) carmencita, ricin purified from *R. communis* cv. zanzibarensis (blue hexagon, solid line) and from *R. communis* cv. sanguineus (red inverted triangle, solid line). Mean curves and standard deviation (SD) of three independent experiments.

**Figure 2 toxins-13-00100-f002:**
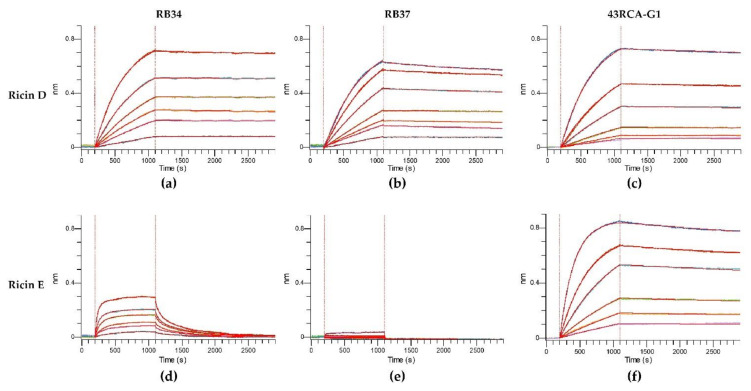
Affinity of antibodies (Abs) for ricin isoforms D and E. Anti-mouse Fc (AMC) sensors were loaded with one of the mouse monoclonal Abs (mAbs) RB34 or RB37 and anti-human Fc (AHC) sensors were loaded with the humanized recombinant Ab (rAb) 43RCA-G1. The loaded sensors were then dipped in wells containing serial dilutions of ricin D or E, followed by immersion in a running buffer. Kinetic curves of affinity for (**a**) RB34/ricin D; (**b**) RB37/ricin D; (**c**) 43RCA-G1/ricin D; (**d**) RB34/ricin E; (**e**) RB37/ricin E; (**f**) 43RCA-G1/ricin E.

**Figure 3 toxins-13-00100-f003:**
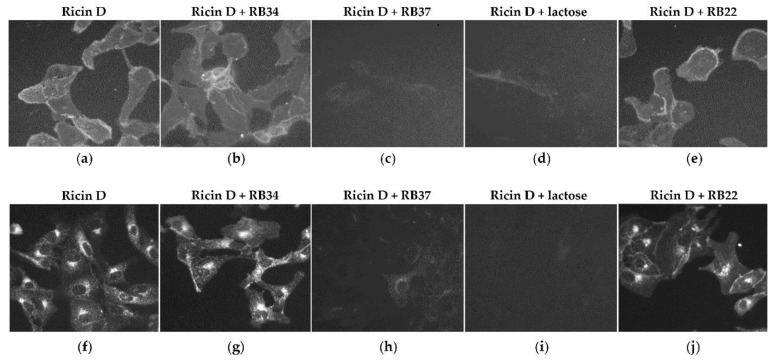
Effect of the mAbs RB34 and RB37 or lactose as control on the binding of ricin to and internalization in Vero cells. Vero cells were incubated with fluorescent ricin D (ricin FP532, 1 µg/mL) pre-incubated with RB34 or RB37 (500 µg/mL). Lactose and a non-neutralizing anti-RTB mAb (RB22) were used as controls. Binding (incubation for 30 min at 4 °C, panels **a** to **e**) and internalization (incubation for 4 h at 37 °C, panels **f** to **j**) of fluorescent ricin D was evaluated through the incubation of cells with ricin alone (panels **a** and **f**) or in presence of RB34 (panels **b** and **g**), RB37 (panels **c** and **h**), lactose (panels **d** and **i**), or the non-neutralizing anti-RTB mAb RB22 (panels **e** and **j**).

**Figure 4 toxins-13-00100-f004:**
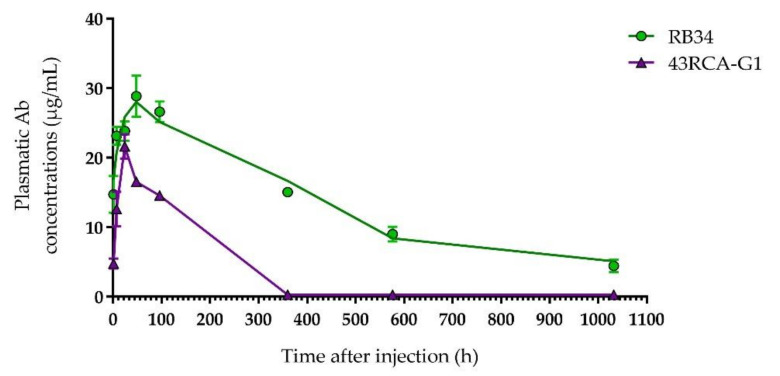
Pharmacokinetic study of RB34 and 43RCA-G1 in mice. Purified Ab (50 µg) was injected by the intraperitoneal (i.p.) route into BALB/cJ mice (*n* = 4 for each time-point). Mice were sacrificed at different times to calculate the plasma concentration of Abs RB34 (green filled circles) and 43RCA-G1 (violet filled triangles) using an immunoassay.

**Figure 5 toxins-13-00100-f005:**
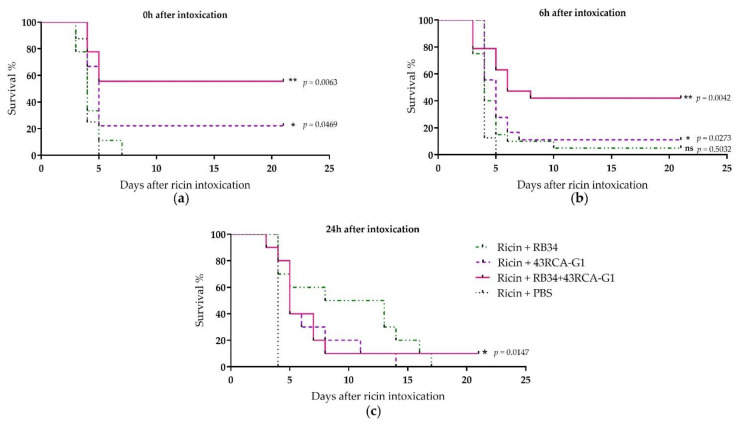
In vivo protective activity of RB34 and 43RCA-G1 Abs alone or combined. Female BALB/cJ mice were instilled with 5 LD_50_ of ricin D+E by the intranasal (i.n.) route and treated by i.p. administration of 10 mg/kg of RB34 (green dashed-dotted line), 43RCA-G1 (violet dashed line), RB34+43RCA-G1 (pink solid line), or PBS (black dotted line) as the control, (**a**) concomitantly (**b**) 6 or (**c**) 24 h after poisoning. Mean curves of two independent experiments (*n* = 10 for each experiment). Statistical analysis: log-rank (Mantel-Cox) tests, ns: not significant, *: *p* < 0.05, **: *p* < 0.01.

**Table 1 toxins-13-00100-t001:** 50% Cytotoxic Dose (CD_50_) of ricin from different *R. communis* cultivars ^#^.

*R. communis* Cultivar	CD_50_ (pg/mL)
Ricin D ^1^	7.4 ± 1.0
Ricin E ^1^	12.8 ± 2.6 *^, ns^
Ricin D+E ^1, †^	8.8 ± 0.8
Carmencita ^2^	8.0 ± 1.3
Zanzibarensis	5.6 ± 0.4
Tanzania	7.5 ± 1.3
Gibsonii	8.2 ± 0.6
Sanguineus	13.4 ± 0.7 *^, ns^
India	10.8 ± 1.7 **
Impala	6.9 ± 1.1
Ricin D ^3^	8.9 ± 0.4
RCA120 ^1^	209.0 ± 26.3 ***

^1^*R. communis* cv. carmencita from French Alternative Energies and Atomic Energy Commission (CEA), ^2^
*R. communis* cv. carmencita from Robert Koch Institute (RKI).^3^
*R. communis* cv. sanguineus gift from Dr. Beaumelle. ^†^ Equimolar solution of ricins D and E. ^#^ Mean CD_50_ values (in pg/mL) and SD of three independent experiments. Statistical analysis: One-way ANOVA with Tukey’s post-hoc test: *: *P* at least <0.05 for ricins E and *R. communis* cv. sanguineus when compared to each of the others with an exception for cv. india; ns: Not significant difference between ricin E and cv. sanguineus; **: *p* < 0.01 for *R. communis* cv. india vs. *R. communis* cv. zanzibarensis; ***: *p* < 0.001 for RCA120 vs. all ricins.

**Table 2 toxins-13-00100-t002:** Kinetic parameters of Ab/Ricin interactions.

Antibody	Ricin Isoform	K_D_ (M) × 10^−11 #^	k_a_ (M^−1^s^−1^) × 10^5 #^	k_dis_ (s^−1^) × 10^−5 #^
RB34	Ricin D	1.0 ± < 0.1 ***	5.61 ± 0.003	0.07 ± 0.01
	Ricin E	542.0 ± 44.3	17.00 ± 0.308	330.00 ± 1.36
RB37	Ricin D	13.7 ± < 0.1	3.25 ± 0.004	4.46 ± 0.01
	Ricin E	ND	ND	ND
43RCA-G1	Ricin D	4.8 ± < 0.1	4.23 ± 0.003	2.00 ± < 0.01
	Ricin E	9.6 ± < 0.1 ***	4.69 ± 0.002	4.52 ± < 0.01

^#^ Values determined by the Octet^®^ Data Analysis HT 10.0 software using a 1:1 binding model, and expressed as mean ± SEM. ND: Not detectable. Student’s t-test performed, ***: *p* < 0.001.

**Table 3 toxins-13-00100-t003:** Concentration of ricin from different *R. communis* cultivars used for the neutralizing assay.

*R. communis* Cultivar	Ricin Concentration (ng/mL)
Ricin D ^1^	0.125
Ricin E ^1^	0.125
Ricin D+E ^1, *^	0.125
Carmencita ^2^	0.125
Zanzibarensis	0.0625
Tanzania	0.0625
Gibsonii	0.125
Sanguineus	0.250
India	0.125
Impala	0.125

^1^ From *R. communis* cv. carmencita from CEA. *Equimolar solution of ricins D and E. ^2^
*R. communis* cv. carmencita from RKI.

**Table 4 toxins-13-00100-t004:** 50% Inhibitory Concentration (IC_50_) of individual Abs or Ab combinations for the inhibition of ricin from different *R. communis* cultivars. ^#^

Antibody/Ricin Cultivar	RB34	RB37	43RCA-G1	RB34+RB37	RB34+ 43RCA-G1	RB37 +43RCA-G1	RB34+RB37 +43RCA-G1
Ricin D ^1^	0.052 ± 0.016 ***	2.024 ± 0.338	41.708 ± 10.209	0.041 ± 0.008	0.022 ± 0.003 ^ns^	0.509 ± 0.050	0.023 ± 0.001
Ricin E ^1^	16.586 ± 2.625	ND	0.392 ± 0.072 **	ND	1.422 ± 0.188 ^ns^	ND	ND
Ricin D+E ^1, †^	4.357 ± 0.919	ND	5.284 ± 0.469	ND	0.351 ± 0.032 ***	1.084 ± 0.234	0.806 ± 0.038
Carmencita ^2^	3.188 ± 0.678	ND	6.985 ± 2.097	nd	0.441 ± 0.092 **	nd	0.550 ± 0.144
Zanzibarensis	0. 011 ± 0.002 *	0.248 ± 0.027	0.615 ± 0.139	0.012 ± 0.002	0.014 ± 0.003 ^ns^	0.145 ± 0.033	0.016 ± 0.002
Tanzania	0.006 ± 0.001 **	0.219 ± 0.048	0.440 ± 0.073	0.008 ± 0.001	0.008 ± 0.002 ^ns^	0.149 ± 0.026	0.009 ± 0.001
Gibsonii	0.337 ± 0.076 ***	3.604 ± 1.077	2.064 ± 0.334	0.258 ± 0.042	0.031 ± 0.005 **	0.299 ± 0.033	0.035 ± 0.004
Sanguineus	2.908 ± 0.458 *	ND	11.338 ± 3.005	nd	0.280 ± 0.052 **	nd	0.435 ± 0.061
India	2.550 ± 0.097	ND	2.701 ± 0.628	nd	0.107 ± 0.026 **	nd	0.210 ± 0.032
Impala	0.052 ±0.012 ***	1.496 ± 0.363	2.777 ± 0.166	0.093 ± 0.020	0.017 ± 0.004 ^ns^	0.219 ± 0.003	0.026 ± 0.005

^#^ Mean IC_50_ values (in µg/mL) ± SD of three independent experiments. ^1^ From *R. communis* cv. carmencita from CEA. ^†^ Equimolar solution of ricins D and E. ^2^
*R. communis* cv. carmencita from RKI; ND: Not detectable; nd: Not determined. Statistical analysis: One-way ANOVA with Tukey’s post-hoc test; ns: Not significant; *: *p* < 0.05, **: *p* < 0.01, ***: *p* < 0.001.

**Table 5 toxins-13-00100-t005:** Kinetic parameters of Ab/ricin or Ab/ricin toxin B-chain (RTB) interactions in the presence of lactose.

Antibody	Ricin Isoform/Chain	K_D_ (M) × 10^−11 #^	k_a_ (M^−1^s^−1^) × 10^5^ ^#^	k_dis_ (s^−1^) × 10^−5^ ^#^
RB34	Ricin D-glucose ^1^	0.73 ± 0.14	4.23 ± 0.011	0.137 ± 0.018
	Ricin D-lactose ^2^	0.14 ± < 0.1 ***	3.17 ± 0.019	0.021 ± 0.013
	RTB-glucose ^1^	0.32 ± < 0.1	3.74 ± 0.004	0.049 ± 0.010
	RTB-lactose ^2^	0.36 ± < 0.1 ^ns^	2.59 ± 0.005	0.054 ± 0.013
RB37	Ricin D-glucose ^1^	16.40 ± 0.58	8.85 ± 0.092	14.30 ± 0.050
	Ricin D-lactose ^2^	120.00 ± 1.25 ***	4.77 ± 0.037	32.90 ± 0.090
	RTB-glucose ^1^	5.52 ± 0.23	4.80 ± 0.028	2.810 ± 0.032
	RTB-lactose ^2^	65.20 ± 0.87 ***	1.69 ± 0.010	8.710 ± 0.048

^1^ Running buffer containing glucose at a 10 mM final concentration. ^2^ Running buffer containing lactose at a 10 mM final concentration. ^#^ Values determined by the Octet^®^ Data Analysis HT 10.0 software using a 1:1 binding model, and expressed as mean ± SEM. Student’s t-test performed, ns: Not significant, ***: *p* < 0.001.

## Data Availability

Data is contained within the article or supplementary material. The data presented in this study are available in Orsini Delgado, M.L.; Avril, A.; Prigent, J.; Dano, J.; Rouaix, A.; Worbs, S.; Dorner, B.G.; Rougeaux, C.; Becher, F.; Fenaille, F.; et al. Ricin Antibodies’ Neutralizing Capacity against Different Ricin Isoforms and Cultivars. Toxins 2021, 13, 100. https://doi.org/10.3390/toxins13020100.

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
