# Peer review of "Ricin Antibodies’ Neutralizing Capacity against Different Ricin Isoforms and Cultivars"

_toxins, 2021, doi:10.3390/toxins13020100_

Round 1
Reviewer 1 Report
The authors provide a comprehensive analysis of the neutralizing ability of a series of anti-ricin antibodies. This comparison evaluated protection from cellular toxicity (calculating CD50), determined binding affinity (Kd, Ka, Kdis), assessed inhibitory capacity of lactose, visualized binding and internalization using fluorescent ricin and Vero cells, and assessed the pharmacokinetics of the antibodies in vivo before determining the protective capability of antibodies alone and in combination. These studies standardize numerous other studies which were performed with different preparations of ricin or which examined individual antibodies. These authors provided an overview which will be useful to the entire field, by standardizing their cultivars for concentration of ricin, comparing ricin D and E, as well as multiple different antibodies alone and in combination. This paper provides insight into the effectiveness of combining antibodies to increase protection against this dangerous bioterrorism agent, while simultaneously creating a powerful resource of other investigators in the field.
The introduction was comprehensive but focused on the key aspects of the field which provided clear rationale for their studies. Their experiments were well described, adequately controlled, statistically validated, and properly interpreted. The discussion provided an in depth review of their conclusions as well as a forward-looking analysis of the implications of their findings. It was a pleasure to read, and will be a useful resource for others in the field.
A few minor technical points: In Figure 2, the differences between the different antibodies would be more apparent if each panel has the same Y axis range.
The inhibitory capability of lactose (and similarly galactose) is impressive. It would be engaging for a non-expert if the authors could provide a bit more discussion about where these sugars are likely to interfere in the binding pathways and the practicality of using (or not using) lactose itself as a therapeutic for ricin toxicity.
Author Response
Dear Reviewer,
We thank you warmly for your very positive feedback, we sincerely appreciate your comments and the suggestions made in order to improve the manuscript.
Point 1: Figure 2, the differences between the different antibodies would be more apparent if each panel has the same Y axis range.
Response 1: As required, we have adjusted the Y axis of the different graphs for all of them to have the same range. It is however important to highlight that for this technique, the signal intensity of the individual curves is not completely correlated to the binding. The kinetic parameters are determined based on a mathematical model that considers the association and dissociation rates (i.e. the shape of the curves, not the intensity), which allow to determine the Ka and Kdis and consequently the KD. Moreover, the sensors used for 43RCA-G1 (anti-human Fc sensor) are different from the sensors used for the other two antibodies (anti-mouse Fc sensor) and no conclusions should be taken based only on the differences of intensity of the signal between the different antibodies.
Point 2: The inhibitory capability of lactose (and similarly galactose) is impressive. It would be engaging for a non-expert if the authors could provide a bit more discussion about where these sugars are likely to interfere in the binding pathways and the practicality of using (or not using) lactose itself as a therapeutic for ricin toxicity.
Response 2: We appreciate your pertinent remark on this point. We effectively used (as other authors) lactose as an inhibitor of ricin binding to the cells, in an in vitro model, in order to evaluate the mechanism of action of the anti-RTB antibodies. These tests allowed us to identify the mechanism of action of RB37 antibody, as a blocker of the binding of ricin to cells. Concerning RB34, the mechanism is much less clear. Our observation that lactose increases the affinity of the antibody is an in vitro observation, we have no experimental evidence that it behaves the same in vivo. However, it can be hypothesized that the binding of ricin to glycoproteins and glycolipids via terminal galactose moieties induces a conformational change of ricin which further increases the binding affinity of the RB34 antibody to ricin, which would already be bound to ricin before it reaches the cell surface. During internalisation of the RB34-ricin complex, ricin would still be bound to galactose. However, it is difficult to know whether this triple bond (antibody-ricin-galactose) is maintained during intracellular transport, as it is not known at what stage of this transport the RB34 antibody plays its role as an inhibitor.
Concerning the practicality of using lactose or galactose as a therapeutic: as sugars, the intravenous administration of galactose or lactose would lead not only to hypergalactosemia, but also to fast metabolization and elimination, probably not compatible with the time window of neutralization of ricin, before it reaches the tissue of interest. Moreover, its administration in huge quantities (10 mM corresponding to 1.8 g/L) would not be compatible with what could be tolerated by the organism. As an osmotically active molecule, the local administration in the lungs might cause an undesirable transfer of water to the lumen causing more inflammation and congestion of the lungs. Finally, the intragastric route (local administration of lactose or galactose) might be of interest; however, one has to consider the fast digestion and absorption of the sugars and the difficulties this might cause in finding an appropriate dose able to reach the tissue beyond the digestive tract. Some authors have evaluated the use of milk [1,2] and suggested that the concentrations of lactose present on milk wouldn’t be enough to protect against an oral exposure to a lethal dose of ricin [2]. Other authors have also evaluated the use of lactose or galactose analogues in vitro in order to find an appropriate inhibitor for the ricin-binding site [3,4], however, none of them was tested in vivo to our knowledge.
In order to justify more clearly the use of lactose in our experiments, we have added a sentence in the section 2.3.1 between lines 221-230. It now reads as follows:
“We have previously shown that RB34 and RB37 recognize ricin chain B (RTB). Since it has previously been described that some anti-RTB antibodies could compete with the galactose binding sites of the B-chain to inhibit ricin entry into cells [27–29], we previously performed a competition assay using RTB, anti-RTB antibodies and lactose as competitor, and we have shown that lactose inhibited the binding of RB37 to RTB, while it enhanced the binding of RB34 to RTB [24]. In order to confirm these data and to understand more deeply the relationship between lactose and the binding of the mAbs to ricin, we evaluated their affinity for ricins D and E and RTB in the absence or presence of lactose, by biolayer interferometry.”
1. Rasooly, R.; He, X.; Friedmans, M. Milk inhibits the biological activity of ricin. J. Biol. Chem. 2012, 287, 27924–27929, doi:10.1074/jbc.M112.362988.
2. Lumor, S.E.; Deen, B.D.; Ronningen, I.; Smith, K.; Fredrickson, N.R.; Diez-Gonzalez, F.; Labuza, T.P. Assessment of the inhibition of ricin toxicity by lactose in milk. J. Food Prot. 2013, 76, 2037–2039, doi:10.4315/0362-028X.JFP-13-091.
3. André, S.; Sansone, F.; Kaltner, H.; Casnati, A.; Kopitz, J.; Gabius, H.J.; Ungaro, R. Calix[n]arene-based glycoclusters: Bioactivity of thiourea-linked galactose/lactose moieties as inhibitors of binding of medically relevant lectins to a glycoprotein and cell-surface glycoconjugates and selectivity among human adhesion/growth-regulatory galectins. ChemBioChem 2008, 9, 1649–1661, doi:10.1002/cbic.200800035.
4. Dawson, R.M.; Alderton, M.R.; Wells, D.; Hartley, P.G. Monovalent and polyvalent carbohydrate inhibitors of ricin binding to a model of the cell-surface receptor. J. Appl. Toxicol. 2006, 26, 247–252, doi:10.1002/jat.1136.
Reviewer 2 Report
Ricin Antibodies neutralizing capacity again different ricin isoforms and cultivars – REVIEW
This paper describes studies done to compare different ricin monoclonal antibodies to different isoforms and cultivars of ricin. It is a well thought out and well present paper, that does add additional knowledge to the current body of knowledge.
My one largest criticism of the paper is that there is no statistical comparisons between any of the studies completed. For example: Are there significant differences in the figure 1 toxicity curves or the CD50 of the ricin in table 1. In my opinion this information should be included in the paper even if there are no statistical difference found. I understand it is important that biological relevance and statistical difference can be different but I believe in this case doing the stats even if they show little difference is am important part of the comparisons made.
Some minor alterations:
Figure 2 - Very difficult to read the axis of the graphs. This figure needs work to make it easier to read
Line 89 - I think you need a citation to support this statement
Line 287 - Typo - Feel should be fell.
Author Response
Dear Reviewer,
We thank you warmly for your very positive feedback, we sincerely appreciate your comments and the suggestions made in order to improve the manuscript.
Point 1: My one largest criticism of the paper is that there is no statistical comparisons between any of the studies completed. For example: Are there significant differences in the figure 1 toxicity curves or the CD50 of the ricin in table 1. In my opinion this information should be included in the paper even if there are no statistical difference found. I understand it is important that biological relevance and statistical difference can be different but I believe in this case doing the stats even if they show little difference is am important part of the comparisons made.
Response 1: We thank you for the highlight. We have added the statistics for the different studies carried out. We have also modified the text accordingly when needed to indicate if there were or not statistically significant differences.
Table 1 has been modified to indicate statistical differences. Footer Table 1, between lines 116-120, now reads: “Mean CD50 values (in pg/mL) and SD of three independent experiments. Statistical analysis: One way ANOVA with Tuckey’s post hoc test: *: p at least < 0.05 for ricin E and R. communis cv. sanguineus when compared to each of the others with exception for cv. india. ns: not significant difference between ricin E and cv. sanguineus. **: p< 0.01 for R. communis cv. india vs R. communis cv. zanzibarensis. ***: p<0.001 for RCA120 vs all ricins.”
Following the information added to Table 1, some modifications have been introduced to the main text between lines 123-145:
“ Ricin isoforms D and E and ricin from the different cultivars show a similar cytotoxicity, with a CD50 of the same order of magnitude (i.e. 10 pg/mL or 0.16 pM), with some differences (Table 1): as previously described, ricin E is less cytotoxic than ricin D [26], with a CD50 that is almost twice that of ricin D (12.8 ± 2.6 pg/mL vs. 7.4 ± 1.0 pg/mL, p<0.001), the cytotoxicity of the equimolar solution of ricins D and E (ricin D+E) being in between (8.8 ± 0.8 pg/mL), though only a statistically significant difference was observed with ricin E (p=0.0078). When comparing to the other cultivars, some of them, including zanzibarensis, which is known to produce only ricin D [17], showed a cytotoxicity
lower though not statistically significant to that of our purified ricin D (Table 1), while other cultivars, previously shown to produce both isoforms, like carmencita, have a cytotoxicity similar to that of ricin D+E (Table 1). The cytotoxicity of ricin D from R. communis cv. sanguineus (“Beaumelle”) is similar to that of the ricin D/E mixture, rather than to ricin D or zanzibarensis, though no statistically significant difference was observed between the CD50 values. It should be noted that ricin purified from carmencita at RKI, which naturally contains a 1:1 ratio of isoforms D and E, presents the same CD50 as the reconstituted D/E mixture from separately purified ricins D and E at the CEA. For other cultivars, containing different proportions of isoforms D and E, cytotoxicities vary from a CD50 close to that of ricin D alone (cultivar Tanzania, 7.5 ± 1.3 pg/mL) to a CD50 close to that of ricin E alone (cultivar sanguineus 13.4 ± 0.7 pg/mL). These differences might correspond to different proportions of ricin D vs. ricin E or differences in activity due to posttranslational modifications and/or purification process. As expected, RCA120 cytotoxicity was at least 10 times lower than that of the different ricins (209.0 ± 26.3 pg/mL, p<0.001).”
Table 2 has been modified to indicate statistical differences. Footer Table 2, between lines 160-161 now reads: “ # Values determined by Octet® Data Analysis HT 10.0 software using a 1:1 binding model, and expressed as mean ± SEM. ND: not detectable. Student’s t-test performed, ***: p<0.001.”
The main text related to Table 2 has been modified between lines 165-176, it now reads: “Altogether, these results show a high affinity of the antibodies for ricin D with an overall KD of 10.0 pM for RB34, 137 pM for RB37 and 48 pM for 43RCA-G1 (Table 2), being RB34 the one with the highest affinity (p<0.001). When evaluating the affinity of the antibodies for ricin E, 43RCA-G1 has an affinity for ricin E similar to that for ricin D (KD = 96 pM, Table 2 and Figure 2f), while the two anti-RTB antibodies (RB34 and RB37) display a very low affinity for ricin E (Table 2, Figure 2d-e) statistically different from 43RCA-G1/ricin E (p<0.001). RB34 has an association rate for ricin E similar to that for ricin D, but a high dissociation rate (Figure 2d), suggesting that the complexes RB34/ricin E are much less stable than the RB34/ricin D ones, giving an overall affinity constant of 5.42 nM, which is 400-fold less than that for ricin D (p<0.001). Concerning RB37, we were unable to calculate a dissociation constant since no association with ricin E was detectable (Figure 2e).”
Table 4 has been modified to indicate statistical differences. Footer Table 4, between lines 209-212, now reads: “ 1 From R. communis cv. carmencita. from CEA. † Equimolar solution of ricins D and E. 2 R. communis cv. carmencita from RKI; ND: not detectable; nd: not determined; # Mean IC50 values (in μg/mL) and SD of three independent experiments. Statistical analysis: One way ANOVA with Tuckey’s post hoc test; ns: not significant, *: p<0.05, **: p<0.01, ***: p<0.001.”
Associated to the modifications in Table 4, some changes have been introduced to the main text between lines 192-206: “Statistical analysis was performed (One way ANOVA with Tuckey’s post hoc test) to compare first the inhibitory capacity of the individual antibodies and then the best individual antibody with the different antibody combinations. All individual antibodies inhibit ricin D activity, RB34 being the most efficient (i.e. lower IC50, p<0.001), which correlates with its kinetic parameters. When combining the antibodies in equal amounts, the RB34+43RCA-G1 combination gives the best protection, and the addition of RB37 does not improve the protecting capacity. Concerning ricin E neutralization, 43RCA-G1 is the best candidate, as expected from kinetic parameters. In this case, we con-firmed the weak affinity of RB34 for ricin E, since the addition of RB34 to 43RCA-G1 is less effective than 43RCA-G1 alone in the in vitro neutralization of ricin E (p<0.01). Finally, when evaluating the neutralization of the equimolar solution of ricins D and E, we observed that RB34 and 43RCA-G1 have a similar neutralizing capacity, RB37 being much less effective. Moreover, combination of RB34 and 43RCA-G1 provides a synergistic neutralizing effect, with an IC50 12 to 15 times lower than the IC50 of each Ab individually (p<0.001).”
And between lines 213-219: “ Concerning the capacity of the Abs to neutralize ricin purified from different cultivars, in most cases, the combination of RB34 and 43RCA-G1 had the best neutralizing capacity, with an IC50 2 to 12 times lower than that of each Ab individually (p<0.01), except for R. communis cv. zanzibarensis, which contains exclusively ricin D [17], and tanzania and impala, for which RB34 showed the best neutralizing capacity, though no statistically significant difference from RB34+43RCA-G1 was observed.”
Table 5 has been modified to indicate statistical differences. Footer Table 5, between lines 241-243, now reads: “ # Values determined by Octet® Data Analysis HT 10.0 software using a 1:1 binding model, and expressed as mean ± SEM. Student’s t-test performed, ns: not significant, ***: p<0.001.”
Concerning the modifications of table 5, some changes have been made to the main text between lines 230-237. It now reads: “In the presence of lactose, the affinity of RB37 for ricin D or RTB is reduced (i.e. KD value increased) due both to a reduction of the association and an increase of the dissociation, giving an overall KD around 10 times higher (p<0.001) (Table 5), confirming competition between lactose and RB37 for the binding site of RTB, whereas the affinity of RB34 for ricin D increases by around 5 times in the presence of lactose (p<0.001), suggesting that the binding of ricin to its ligand induces a conformational change in holo-ricin D that enhances the binding, yielding a more stable RB34/ricin D complex (Table 5).
The sub-section 4.7. of materials and methods was modified between lines 582-588 as follows: “All cell viability assays, kinetic parameter assays and in vivo treatment assays were performed at least twice. Statistical analysis for cell viability assay and in vivo experiments, was performed with PRISM® software v.5.04 (GraphPad Software inc., San Diego, CA). For cell viability assays, a non-linear regression 4-parameter variable slope test was used, and one-way ANOVA with Tukey’s post-hoc test performed. For survival curves of in vivo experiments, log-rank (Mantel-Cox) tests were used. For the kinetic parameter assays a Student t-test was performed. Data is represented as means ± SEM.”
Point 2: Figure 2 - Very difficult to read the axis of the graphs. This figure needs work to make it easier to read.
Response 2: Figure 2 has been improved to show the same range in the Y axis and to make it easier to read, unfortunately the software does not allow to modify the font size, which limits the possibility of improving even more the figure. We hope that this will still be satisfactory and sufficient.
Point 3: Line 89 - I think you need a citation to support this statement.
Response 3: As requested, citations have been added to the statement in line 89.
Point 4: Line 287 - Typo - Feel should be fell.
Response 4: We are sorry for this spelling mistake in line 287 (now line 309) which is now corrected.